# Expression Profiles of Long Non-Coding RNAs in the Articular Cartilage of Rats Exposed to T-2 Toxin

**DOI:** 10.3390/ijms241813703

**Published:** 2023-09-05

**Authors:** Fangfang Yu, Miao Wang, Kangting Luo, Lei Sun, Shuiyuan Yu, Juan Zuo, Yanjie Wang

**Affiliations:** School of Public Health, Zhengzhou University, Zhengzhou 450001, China; yufangfang@zzu.edu.cn (F.Y.); wangmiao_9842@163.com (M.W.); lkt1307@163.com (K.L.); sl349496017@163.com (L.S.); ysy599926@163.com (S.Y.); zuojuan719920@163.com (J.Z.)

**Keywords:** T-2 toxin, articular cartilage, autophagy, lncRNAs, RNA-Seq

## Abstract

T-2 toxin could induce bone damage. But there is no specific mechanism about the long non-coding RNAs (lncRNAs) involved in T-2 toxin-induced articular cartilage injury. In this study, 24 SD rats were randomly divided into a control group and a T-2 group, which were administered 4% absolute ethanol and 100 ng/g · bw/day of T-2 toxin, respectively. After treatment for 4 weeks, safranin O/fast green staining identified the pathological changes in the articular cartilage of rats, and immunofluorescence verified the autophagy level increase in the T-2 group. Total RNA was isolated, and high-throughput sequencing was performed. A total of 620 differentially expressed lncRNAs (DE-lncRNAs) were identified, and 326 target genes were predicted. Enrichment analyses showed that the target genes of DE-lncRNAs were enriched in the autophagy-related biological processes and pathways. According to the autophagy database, a total of 23 autophagy-related genes were identified, and five hub genes (Foxo3, Foxo1, Stk11, Hdac4, and Rela) were screened using the Maximal Clique Centrality algorithm. The Human Protein Atlas database indicated that Rela and Hdac4 proteins were highly expressed in the bone marrow tissue, while Foxo3, Foxo1, and Stk11 proteins were reduced. According to Enrichr, etoposide and diatrizoic acid were identified as the key drugs. The real-time quantitative PCR results were consistent with the RNA sequencing (RNA-Seq) results. These results suggested that autophagy was involved in the rat articular cartilage lesions induced by T-2 toxin. The lncRNAs of NONRATG014223.2, NONRATG012484.2, NONRATG021591.2, NONRATG024691.2, and NONRATG002808.2, and their target genes of Foxo3, Foxo1, Stk11, Hdac4, and Rela, respectively, were the key regulator factors of autophagy.

## 1. Introduction

T-2 toxin is a kind of trichothecene mycotoxin produced by various Fusarium species and is known as the most toxic fungal secondary metabolite [1]. It mainly contaminates wheat, barley, maize, oats, and corn, which are closely related to human diet [2]. And it could induce oxidative stress [3], inflammatory response [4], apoptosis, and the autophagy of chondrocytes [5]. This autophagy is critical for the formation and development of cartilage [6]. It is a self-degradative process that is regulated by a series of autophagy-related genes (ATGs), and it could eliminate damaged organelles and maintain intracellular homeostasis [7,8]. Autophagy can be divided into four processes: (1) the activation of autophagy: the ATG1/ULK1-containing complex is involved in this process; (2) autophagosome formation: PI3K-ATG14, ATG18, and the ATG5-ATG12-ATG16L1 complex is involved in autophagosome expansion and maturation as well as in ATG3, ATG4, and ATG7-mediated lipidation of LC3; (3) the transport and fusion of autophagosomes with lysosomes; (4) degradation and recycling: hydrolases degrade the autophagosome, release degradation products into the cytosol, and then recycle them [9]. T-2 toxin is widely recognized as an important etiology of the Kashin–Beck disease (KBD) [10,11]. It is characterized by deep necrosis of the epiphysis plate and articular cartilage with a higher rate of disability, which greatly burdens the diseased person’s family and society [12,13]. Moreover, T-2 toxin also promotes proteoglycans degradation and activates catabolism via the ROS-NF-κB-HIF-2α pathway in the chondrocytes [14,15]. Therefore, it is important to identify the potential mechanism of T-2 toxin-induced articular cartilage damage.

Long non-coding RNAs (lncRNAs) are a class of non-coding RNAs longer than 200 nucleotides, which transcribe under the direction of RNA polymerase II/III [16]. LncRNAs account for a large proportion of the non-coding transcriptome [17]. Since lncRNAs cannot encode any proteins, they are generally considered as transcriptional “noise” by the scientific community [18]. However, increasing evidence suggests that lncRNAs play an important role in chromatin structure and function regulations, genetic transcription modulation, and RNA metabolism, as well as in cell autophagy, apoptosis, and embryonic development [19,20]. Previous studies demonstrated that lncRNAs are involved in many human diseases, including osteoarthritis and KBD [21]. For example, lncRNA HAGLR mediated the chondrocytes’ inflammatory response through the miR-130a-3p/JAK1 axis [22], and lncRNA HOTAIR accelerated mechanical-stimulation-induced apoptosis via regulating the miR-221/BBC3 axis in chondrocytes [23]. Furthermore, lncRNAs could also facilitate extracellular matrix synthesis by interacting with RNA-binding protein and activating the PI3K/AKT pathway [24,25]. However, less research has been focused on lncRNAs’ expression profiles’ change in rat articular cartilage induced by T-2 toxin.

In the current study, safranin O/fast green staining was used to measure the pathological changes in articular cartilage. The lncRNA profiles of SD rat articular cartilage in the T-2 group and the control group were obtained using high-throughput sequencing technology. Then, the target genes of differentially expressed lncRNAs (DE-lncRNAs) were predicted, and gene ontology (GO) and Kyoto Encyclopedia of Genes Genomes database (KEGG) enrichment analyses were performed. Immunofluorescence was performed to indicate the change in the autophagy level of the articular cartilage in the T-2 group. Consequently, a protein–protein interaction (PPI) network was constructed for screening the hub genes, the Human Protein Atlas (HPA) database was used to validate the change of hub proteins, and a real-time quantitative polymerase chain reaction (RT-qPCR) was used to validate the results of RNA sequencing (RNA-Seq).

## 2. Results

### 2.1. Safranin O/Fast Green Staining of Articular Cartilage

The safranin O/fast green staining of the articular cartilage in the T-2 group and the control group is shown in Figure 1. In the control group, the cartilage matrix was deep red, and the chondrocytes were closely arranged in a column. After being administered T-2 toxin for 4 weeks, the cartilage matrix was lightly stained, and the range of the empty lacunae increased, which suggested proteoglycan loss and chondrocytes death, respectively. Moreover, an irregular arrangement of chondrocytes can be observed in the T-2 group compared with the control group.

### 2.2. Quality Control of RNA Sequencing Data and Read Mapping

A total of 58.10 million reads and 22,127 lncRNAs were obtained. After quality control, we obtained high-quality reads with Q30 > 95%, clean reads% > 99.0%, and the total bases of R1 plus R2 ranged from 12.6 to 14.1 Gb (Table 1). The data achieved the quality control standard with Q30 > 80%, clean reads% > 90.0%, and the total bases of R1 plus R2 were >12 Gb in 90% of the samples, which suggested the good quality of the RNA-Seq. After these reads were mapped to the reference database, the rate was from 89.7% to 90.8%. Taken together, these data indicated that the sequencing quality was good and sufficient for subsequent analysis.

### 2.3. Identification of Differentially Expressed lncRNAs

According to *p*-value < 0.05 and | log_2_ (fold change) | > 1, a total of 620 DE-lncRNAs were identified in the T-2 group compared with the control group, of which 360 were upregulated and 260 were downregulated. The volcano plot shows the distribution of DE-lncRNAs (Figure 2A). In addition, Figure 2B shows a hierarchical cluster analysis of DE-lncRNAs in the form of a heatmap, and the result showed that three samples (i.e., T-2_1, T-2_2, and T-2_3) from the T-2 group were clustered in the same cluster, while three samples (i.e., control_1, control_2, and control_3) from the control group were clustered in the other cluster. Significant differences were observed between the T-2 group and the control group, thereby validating the reliability of our sequencing data.

### 2.4. Functional Enrichment Analyses of Target Genes in the Cartilage Tissues Samples

According to minimum free energy, 326 target genes of DE-lncRNAs were predicted. Then, GO and KEGG analyses were performed on these genes. According to the gene ratio, the top 10 biological process (BP), cellular component (CC), and molecular function (MF) terms are presented in Figure 3A. BP terms were found to be significantly enriched in the regulation of actin filament polymerization, regulation of actin filament length, as well as bone development. CC terms were found to be enriched in the actin filament, autophagosome, cortical actin cytoskeleton, and mitochondrial outer membrane. Moreover, MF terms were related to protein kinase regulator activity, kinase regulator activity, and microtubule binding. Next, we performed KEGG analysis, and the top 20 KEGG pathways of the target genes are depicted in Figure 3B. The pathways were found to be significantly enriched in autophagy-animal, endocytosis, mitophagy-animal, the AMPK signaling pathway, and the FoxO signaling pathway.

### 2.5. T-2 Toxin Triggered Autophagy in Articular Cartilage

As shown in Figure 4, immunofluorescence assay results demonstrated that the LC3 protein level of articular cartilage increased in the T-2 group compared with the control group. These results suggested that the autophagy level was highly expressed in the articular cartilage administered with T-2 toxin.

### 2.6. PPI Network Analysis and the Validation of Hub Genes

A total of 23 ATGs were mapped according to the autophagy database. Then, they were imported in STRING, and the results were presented using Cytoscape. As shown in Figure 5A, the PPI network of the target genes consisted of 14 nodes and 16 edges. Additionally, the top five hub genes were ranked according to the Maximal Clique Centrality (MCC) algorithm (Figure 5B): Foxo3, Foxo1, Hdac4, Rela, and Stk11. The top five hub proteins were then screened using the HPA database and the results are shown in Figure 5C. The Rela and Hdac4 proteins were highly expressed in the bone marrow tissue, whereas the levels of Foxo3, Foxo1, and Stk11 reduced in the bone marrow tissue.

### 2.7. Prediction of Candidate Drugs

The potential intervention drugs of T-2 toxin-induced autophagy were predicted via the Enrichr platform. The top 10 candidate drugs are listed in Table 2. The results showed that etoposide (CTD 00005948) and diatrizoic acid (CTD 00005787) were the two key drugs.

### 2.8. Validation of DE-lncRNAs via RT-qPCR

To further validate the results of RNA-Seq, the lncRNAs of NORATG013414.2 and NONRATG026881.1 were randomly selected for RT-qPCR analysis. These results showed that the relative expression levels of NONRATG026881.1 and NONRATG013414.2 were all upregulated in the T-2 group compared with the control group (Figure 6A,B). According to log_2_ (fold change), the expression trends were consistent with the RNA-Seq results (Figure 6C).

## 3. Discussion

It is well known that T-2 toxin is harmful to human health and causes cytotoxicity, immunotoxicity, and skeletal toxicity, among other harmful effects [26]. However, one of the most severe consequences of T-2 toxin exposure is articular cartilage damage [27]. However, the function of lncRNAs in the T-2 toxin-induced cartilage damage remains unclear. Therefore, this study investigated the change in lncRNAs’ expression profiles in rat articular cartilage using RNA-Seq. Subsequently, we performed functional enrichment analyses of DE-lncRNAs’ target genes. These results suggested that autophagy is a key mechanism for T-2 toxin-induced cartilage damage. 

In this study, safranin O/fast green staining revealed the pathological changes in the rat articular cartilage samples that were exposed to T-2 toxin for 4 weeks. In total, we identified 620 DE-lncRNAs and predicted 326 target genes according to the articular cartilage profile. The 326 identified target genes were then subjected to enrichment analyses. According to the GO analysis, the target genes were mainly enriched in the negative regulation of cytoskeleton organization, regulation of actin filament polymerization, regulation of actin polymerization or depolymerization, and bone development process. As we all know, actin is the most abundant protein that participates in protein–protein interactions in numerous eukaryotic cells, and studies have reported that the balance between polymerization and depolymerization of actin is crucial for therapeutically controlling the chondrocyte phenotype [28]. Next, the KEGG pathway enrichment analysis result showed that autophagy-animal, endocytosis, mitophagy-animal, the AMPK signaling pathway, and the FoxO signaling pathway were significantly enriched. Autophagy is a process involving lysosome degradation under various stress conditions, such as DNA damage [29], toxin exposure [30], nutrient deficiency [31], and hypoxia [32]. Autophagy has been further divided into microautophagy, chaperone-mediated autophagy, and macroautophagy. LC3-associated endocytosis is regarded as a noncanonical autophagy pathway [33]. Moreover, previous studies have found that T-2 toxin exposure can induce an autophagic response that is protective in nature [34]. Thus, moderate autophagy could relieve T-2 toxin-induced toxicity by removing damaged mitochondria [35]. However, excessive activation of autophagy may lead to the degradation of basic cellular components, thereby leading to cell death, further inducing the formation of articular cartilage lesions [36]. In the process of mitophagy, PINK1 recruits and activates Parkin on the surface of dysfunctional mitochondria in response to oxidative stress. Then, Parkin ubiquitinates the proteins on the outer surface of the mitochondrial membrane. Fusion of autophagosome and lysosome can degrade the damaged mitochondrion, further relieving the damages induced by T-2 toxin exposure [37]. AMPK is a serine/threonine protein kinase that acts as an energy sensor in cells. Under pathological conditions, AMPK is phosphorylated and activated, thereby promoting autophagy via ATGs (i.e., ULK1, PIK3C3/VPS34, and mTORC1). Moreover, it can inhibit mitophagy via the regulation of transcription factors located downstream of Foxo3 [38]. Similarly, Foxo induces autophagy via transactivation of autophagy genes and regulation of autophagy activity. Post-translational modification and epigenetic modulation of Foxo can induce or inhibit autophagy [39,40]. Our KEGG enrichment result substantiated autophagy is involved in articular cartilage lesions exposed to T-2 toxin. Furthermore, immunofluorescence assay verified that the level of autophagy in articular cartilage was higher in the T-2 group relative to the control group.

PPI network identified five hub ATGs—Foxo3, Foxo1, Hdac4, Stk11, and Rela—which play important roles in autophagy. These ATGs have a variety of functions; for instance, they can deliver extracellular cargo to the lysosome, promote localization to the plasma membrane and trigger extracellular release of intracellular cargo, and coordinate with various other cell signaling pathways [8]. Foxo3 and Foxo1, as members of the Foxo family, induce autophagy by directly interacting with autophagy proteins [40]. In our study, Foxo3 and Foxo1 were regulated by the lncRNAs of NONRATG014223.2 and NONRATG012484.2, respectively. Foxo3 can activate the transcription of ULK1 and PIK3CA, which further activates the initiation of autophagy [41]. Under stress, Foxo1 is acetylated and binds to ATG7, thereby driving classical degradative autophagy via ATG8 lipidation or LC3-associated phagocytosis [42,43]. Studies have also demonstrated that target regulating Foxo1 in chondrocytes can prevent osteoarthritis via autophagy [44]. Hdac4 is a type of histone deacetylase located in the nucleus and cytoplasm [45] that we found to be regulated by lncRNA NONRATG024691.2. Moreover, Hdac4 can activate Foxo3 that subsequently interacts with ATG5 and LC3II, further inducing autophagy [46]. Stk11, also named LKB1, is a serine/threonine kinase that is widely expressed in tissues. In our study, Stk11 was found to be regulated by NONRATG021591.2. Stk11 activates the phosphorylation of the threonine 172 site on the AMPK α subunit. Next, AMPK-induced autophagy proceeds by inhibiting the mTOR signaling pathway and triggering ATG5 activity [47,48]. Rela (p65), an NF-κB subunit, always activates the NF-κB signaling pathway [49]. NF-κΒ triggers autophagy by directly inducing the expression of Beclin1, ATG5, and LC3 [50]. We found that Rela was regulated by NONRATG002808.2, and its specific regulatory mechanism is shown in Figure 7. Furthermore, HPA database indicated that the proteins of Hdac4 and Rela were highly expressed in the bone marrow tissue, but proteins of Foxo3, Foxo1, and Stk11 were reduced. 

Next, we used the hub ATGs to predict potential drugs. The results suggested that etoposide and diatrizoic acid are the important pharmacological targets. Etoposide is a non-alkaloid lignan that is isolated from the dried roots and rhizomes of *Podophyllum peltatum* or *Podophyllum emodi.* It is used for the treatment of numerous cancer types [51,52]. Additionally, etoposide is also a derivative of podophyllotoxin that exhibits strong anti-inflammation and anti-oxidative activities [53]. Therefore, etoposide could be used to help scavenge the reactive oxygen and intervene in the autophagy response, thereby relieving T-2 toxin-induced cartilage tissue damage. Diatrizoic acid is an iodinated aromatic that is often used as a radiocontrast agent, and it can also induce mitophagy and oxidative stress via calcium dysregulation [54]. Studies have demonstrated that diatrizoic acid is a component of FocusClear™, which has been used for optical clearing of mouse brain, skin, bone, and lymph node cortex tissues [55,56]. Moreover, oral meglumine diatrizoate esophagography could screen esophageal fistulas during radiotherapy or chemoradiotherapy with minimal side effects [57]. Therefore, we speculated that diatrizoic acid could also be used for treating articular cartilage damage. 

However, our study has certain limitations. First, an animal model was used to analyze DE-lncRNAs in the articular cartilage of rats exposed to T-2 toxin, but this was not tested in humans in a clinical setting. Second, the specific regulatory mechanisms between the lncRNAs and their target genes were not established during the experimental exploration. Further molecular biology studies are required to verify the regulatory mechanisms involved.

## 4. Materials and Methods

### 4.1. Establishment of Animal Models Treated with T-2 Toxin 

Twenty-four specific-pathogen-free male Sprague–Dawley (SD) rats, weighing approximately 60–80 g at the age of 4 weeks, were purchased from the Henan provincial laboratory animal center. Rats were adaptively fed in a room with a humidity of 55% ± 5% and a temperature of 22 °C ± 2 °C under a 12 h light/dark cycle for 1 week [27,58]. Then, the rats were weighed and randomly divided into two groups: the control group rats were administered 4% absolute ethanol via gavage and the T-2 rats were administered 100 ng/g · bw/day T-2 toxin via gavage [27,59]. After 4 weeks of treatment, the articular cartilages of the twelve rats in each group were collected. The animal experiments were reviewed and approved by the Animal Ethics Research Committee of Zhengzhou University (ZZUIRB2021-69).

### 4.2. Safranin O/Fast Green Staining

Three articular cartilage samples from each group were fixed in 4% paraformaldehyde for 24 h and decalcified in EDTA decalcifying solution for 4 weeks. Next, each sample was embedded, sliced, and dewaxed to generate cartilage tissue sections. These sections were stained with Weigert solution and rinsed with water before placing in 0.2% fast green solution for 5 min, 1% ethylic acid solution for 15 s, and 0.1% safranin O solution for 5 min. Then, the samples were dehydrated, transparentized, and mounted with neutral balsam. The pathological changes in the articular cartilage between the T-2 group and the control group were scanned using a scanner (3DHISTECH, Budapest, Öv u. 3., Hungary).

### 4.3. Library Construction and RNA-Seq

For RNA-Seq, we first isolated the total RNA from the articular cartilage of three rats from each group using TRIzol (Thermo Fisher Scientific, Waltham, MA, USA). Qubit 2.0 fluorometer (Invitrogen, Carlsbad, CA, USA) was used to quantify the concentration of total RNA (sample concentration ≥ 100 ng/µL, total ≥ 2 μg), NanoDrop 2000 (Thermo Fisher Scientific, Waltham, MA, USA) was used to measure the RNA purity with OD_260/280_ in the range of 1.8–2.2 and OD_260/230_ ≥ 2.0. The RNA degradation was analyzed using agarose gel electrophoresis, and Agilent 2100 Bioanalyzer (Agilent Technologies, Santa Clara, CA, USA) was used to evaluate the RNA integrity (RNA integrity number ≥ 7). 

To ensure the construction of high-quality RNA-Seq libraries, moderate amounts of RNA were sampled for ribosomal RNA (rRNA) depletion. Subsequently, RNAs were purified and digested into 100–300 bp fragments. The first complementary DNA (cDNA) chain was synthesized with short fragment primers and followed by the second cDNA chain synthesis. The 3′-ends of the resulting double-stranded cDNA were adenylated and ligated to Illumina sequencing adaptors using T4 DNA ligase. Subsequently, the initial library was purified using an Agencourt SPRIselect reagent kit (Beckman Coulter, Miami, FL, USA). The library with a peak fragment size of 300 bp was selected and then amplified using PCR. After amplification, a Qubit 2.0 (Invitrogen, Carlsbad, CA, USA) was used to quantify the DNA concentration of the constructed libraries. In addition, an Agilent 2100 Bioanalyzer (Agilent Technologies, Santa Clara, CA, USA) was used to determine the size distribution of the library fragments. After quality inspection, an Illumina HiSeq2500 (Illumina, San Diego, CA, USA) was used to sequence the constructed libraries via a 2 × 150 bp paired sequencing strategy. 

### 4.4. Identification of Differentially Expressed lncRNAs 

FastQC software v0.12.1 (http://www.bioinformatics.babraham.ac.uk/projects/fastqc, accessed on 13 August 2022) was used to filter out the adaptor and low-quality reads. The filter criteria as follows: (1) sequencing primers, (2) low quality sequences near the terminal (Q < 15), and (3) sequences with fragment length < 40 bp. Then, the high-quality clean reads were obtained and were mapped to the reference database using STAR software v2.7.11a (https://github.com/alexdobin/STAR, accessed on 13 August 2022. The results were subsequently analyzed using Picard v3.1.0 (http://broadinstitute.github.io/picard/, accessed on 13 August 2022). The expression levels of these lncRNAs were calculated based on fragments per kilobase of genes per million fragments mapped (FPKM) values. DE-lncRNAs were defined as *p*-value < 0.05 and | log_2_ (fold change) | > 1 via the Deseq2 software v1.40.2 (https://bioconductor.org/packages/release/bioc/html/DESeq2.html, accessed on 13 August 2022) [60]. Hierarchical cluster analysis was also used to determine the expression patterns of DE-lncRNAs.

### 4.5. Target Gene Prediction

The target genes of identified DE-lncRNAs were predicted using lncTar software v1.0 (http://www.cuilab.cn/lnctar, accessed on 13 August 2022). In this project, the minimum free energy of lnRNAs and mRNA binding sites were calculated to determine whether they could produce stable binding sites, and the default threshold of minimum free energy was −0.1.

### 4.6. GO and KEGG Pathway Enrichment Analyses

The target genes of identified DE-lncRNAs were subjected to GO and KEGG enrichment analyses using R software v4.3.1 (https://mirrors.tuna.tsinghua.edu.cn/CRAN/, accessed on 12 May 2023) cluster Profiler that annotates the functions of genes. In GO analysis, the vocabularies included the BP, CC, and MF. In KEGG analysis, the enrichment scores were used to rank the pathways and the pathways with *p*-value < 0.05 were considered significant. 

### 4.7. Immunofluorescence Assay

The paraffin tissues of articular cartilage were sectioned, dewaxed in xylene, and dehydrated using gradient ethanol. After antigen retrieval, the sections were incubated with 3% BSA for 30 min. Then, they were incubated with primary antibodies against microtubule-associated protein 1 light chain 3 alpha (LC3, 1:400, 14600-1-AP, Proteintech, Wuhan, China) at 4 °C overnight, followed by rinsing the sections thrice in PBS solution and incubation with Cy3-conjugated secondary antibodies. Next, the DAPI staining solution was utilized to stain the nucleus of the cell. Immunofluorescence images were captured, and the fluorescence images were quantified using Image-Pro Plus 6.0 (Media Cybernetics, Rockville, MD, USA).

### 4.8. PPI Network Construction and Validation of Hub Genes

The autophagy database (http://www.tanpaku.org/autophagy/, accessed on 12 May 2023) is a manually compiled and managed database that provides up-to-date information regarding autophagy-related proteins and their homologs in 41 eukaryote species [61]. In the current study, it was used to map the autophagy-related target genes. The STRING database (https://cn.string-db.org/, accessed on 12 May 2023) was used to analyze the information regarding interactions among these mapped genes. These interactions were reflected by the comprehensive score. With a confidence score (0.4) set as the minimum interaction score, the autophagy-related target genes were used to construct a PPI network [62]. These results were presented and analyzed using Cytoscape v3.9.0 (https://cytoscape.org/, accessed on 12 May 2023) [63]. Next, we used cyto-Hubba, a Cytoscape plugin, to screen the top five hub genes via MCC [64]. 

The HPA database (https://www.proteinatlas.org/, accessed on 13 May 2023) consists of cell, tissues, and pathology atlas that provides the information regarding transcriptome and protein mapping for specific human tissues. In the current study, HPA database was used to obtain the data on hub protein expression levels in the bone marrow and skeletal muscle tissues.

### 4.9. Prediction of Candidate Pharmacological Targets

The identified five hub genes were applied to predict the candidate pharmacological targets using Enrichr v3.2.0 (https://maayanlab.cloud/Enrichr/, accessed on 13 May 2023). Enrichr is a comprehensive search engine that integrates biological knowledge with a comprehensive database of curated gene sets from previous studies [65]. Currently, Enrichr contains 211 gene set libraries, 431,749 terms, and 60,534,815 gene sets for analysis. Notably, it is important to evaluate protein–drug interactions to understand the structural features required for receptor sensitivity. The candidate pharmacological targets of the five hub genes were sorted by *p*-value from small to large, and the *p*-value of < 0.05 was considered statistically significant [66]. 

### 4.10. Verification Analysis

Total RNA from cartilage tissue was reverse-transcribed into cDNA using SuperScript IV reverse transcriptase (Thermo Fisher Scientific, Waltham, MA, USA). Two lncRNAs were then randomly selected for RT-qPCR using SYBR green (Gene Copoeia Inc., Rockville, MD, USA). GAPDH was used as an internal reference gene. The experimental conditions of the PCR amplification were set as follows: 94 °C for 5 min, followed by 40 cycles at 94 °C for 10 s, 55 °C for 20 s, and 72 °C for 20 s. The lncRNAs’ expression levels were calculated with the 2^−ΔΔCt^ method. Three replicates were used for each group.

## 5. Conclusions

Autophagy was involved in the process of T-2 toxin-induced articular cartilage lesions. The lncRNAs of NONRATG014223.2, NONRATG012484.2, NONRATG021591.2, NONRATG024691.2, and NONRATG002808.2, and their target genes of Foxo3, Foxo1, Stk11, Hdac4, and Rela, respectively, were the key regulatory factors of autophagy.

## Figures and Tables

**Figure 1 ijms-24-13703-f001:**
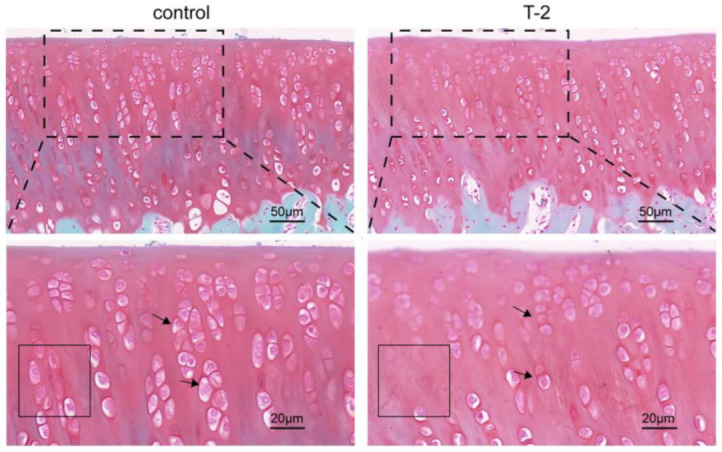
The safranin O/fast green staining of articular cartilage in the T-2 group and the control group; the cartilage tissue was dyed red and the bone tissue was dyed green. The black dashed box represents partially enlarged range; the black solid box represents the empty lacunae increased range. The arrows represent the chondrocytes.

**Figure 2 ijms-24-13703-f002:**
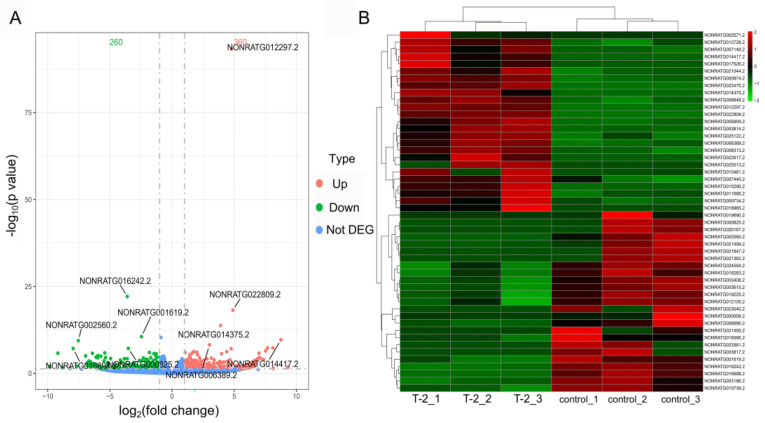
Differentially expressed lncRNAs (DE-lncRNAs) and hierarchical cluster analysis. (**A**) Volcano plot of DE-lncRNAs in the T-2 group compared with the control group. (**B**) Heatmap of DE-lncRNAs in the T-2 group and the control group. Red represents upregulated lncRNAs, green represents downregulated lncRNAs, and blue represents not differentially expressed lncRNAs. The dotted line represents the threshold of DE-lncRNAs with *p*-value < 0.05 and | log_2_ (fold change) | > 1, *n* = 3. The color scale-bar from green to red indicates the increased expression levels of lncRNAs.

**Figure 3 ijms-24-13703-f003:**
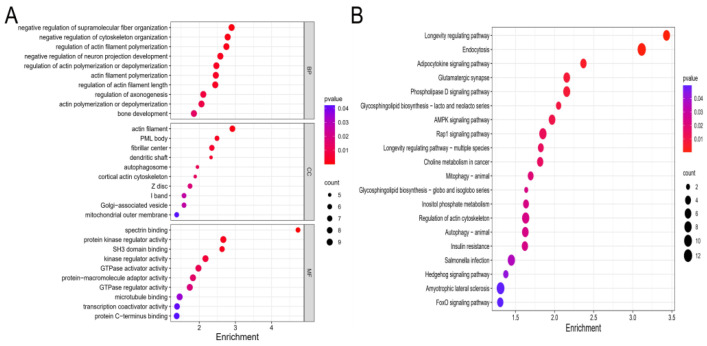
GO and KEGG pathway enrichment analyses. (**A**) The top 10 biological process (BP), molecular function (CC), and molecular function (MF) terms of the target genes in the T-2 group vs. the control group. (**B**) The top 20 enriched pathways of the target genes in the T-2 group compared with the control group. Pathways with *p*-value < 0.05 were considered significant. Enrichment represents -log_10_ (*p*-value).

**Figure 4 ijms-24-13703-f004:**
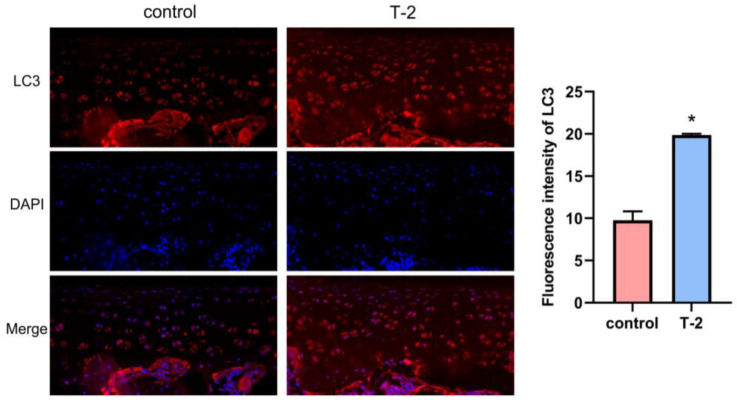
Immunofluorescence staining of LC3 protein in the articular cartilage. Red represents the LC3 protein and blue represents DAPI. Scale bar is 20 μm. Data are represented as mean ± SEM (*n* = 3), * *p*-value < 0.05.

**Figure 5 ijms-24-13703-f005:**
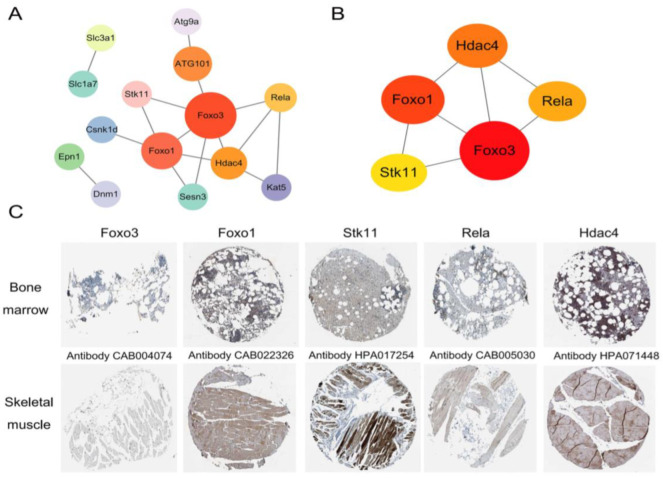
Construction of PPI network and the validation of hub genes. (**A**) The PPI network among the autophagy-related target genes was constructed using STRING database and presented via Cytoscape. (**B**) The top five hub genes identified via the Maximal Clique Centrality algorithm. (**C**) Immunohistochemistry images of the five hub proteins in the bone marrow and skeletal muscle tissues from the Human Protein Atlas database. Scale bar is 200 μm.

**Figure 6 ijms-24-13703-f006:**
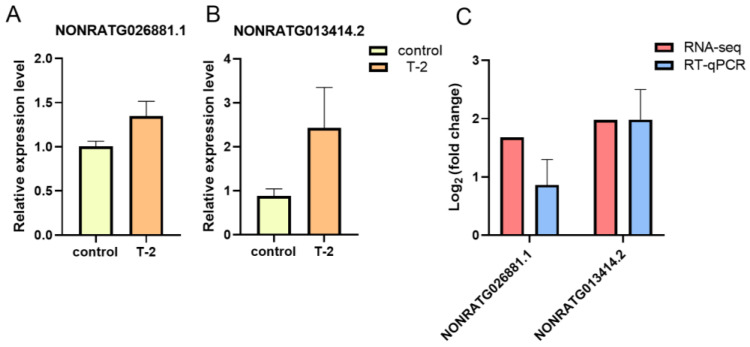
Validation of RNA-Seq results using real-time quantitative PCR(RT-qPCR). The relative expression levels of NONRATG026881.1 (**A**) and NONRATG013414.2 (**B**) in the T-2 group vs. the control group. (**C**) The comparison of two DE-lncRNAs between RNA-Seq and RT-qPCR. Data are represented as mean ± SEM (*n* = 3).

**Figure 7 ijms-24-13703-f007:**
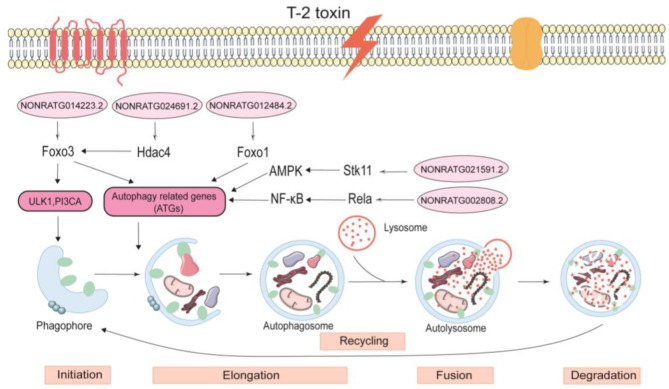
The specific autophagy regulatory mechanism of articular cartilage induced by T-2 toxin. The lncRNA NONRATG014223.2 regulated Foxo3, activating ULK1, PI3CA-dependent autophagy. The lncRNAs of NONRATG024691.2, and NONRATG012484.2 severally regulated the target genes of Hdac4 and Foxo1, driving autophagy through acting on ATGs, respectively. The lncRNAs of NONRATG021591.2 and NONRATG002808.2 regulated target genes of Stk11 and Rela, and then acting on AMPK and NF-κB signaling to activate autophagy, respectively.

**Table 1 ijms-24-13703-t001:** The total quality and mapped results of RNA-Seq.

Samples	Raw Data			Clean Data			Clean Reads (%)	Total Mapped Reads (%)
	Reads	Total Raw Bases (Gb)	Q30 (%)	Reads	Total Clean Bases (Gb)	Q30 (%)		
T-2_1	47,053,795	13.0	95.1	46,831,502	12.9	95.4	99.5	90.4
T-2_2	50,495,169	14.0	94.9	50,276,942	14.0	95.1	99.6	90.8
T-2_3	51,064,323	14.2	95.1	50,818,840	14.1	95.4	99.5	90.7
control_1	49,245,939	13.7	95.5	49,020,806	13.6	95.7	99.5	90.2
control_2	47,073,831	13.1	95.4	46,802,405	13.0	95.7	99.4	89.7
control_3	45,589,580	12.6	95.4	45,383,773	12.6	95.5	99.5	90.0

**Table 2 ijms-24-13703-t002:** Prediction of the top 10 candidate drugs.

Name of Drugs	*p*-Value	Adjusted *p*-Value	Genes
etoposide CTD 00005948	1.37 × 10^−6^	7.65 × 10^−4^	HDAC4, FOXO3, FOXO1, RELA
diatrizoic acid CTD 00005787	2.75 × 10^−6^	7.68 × 10^−4^	FOXO3, RELA
mollugin CTD 00002058	7.64 × 10^−6^	7.99 × 10^−4^	STK11, RELA
metformin BOSS	8.19 × 10^−6^	7.99 × 10^−4^	STK11, FOXO3, FOXO1
2-Butanone BOSS	9.14 × 10^−6^	7.99 × 10^−4^	STK11, FOXO3, FOXO1
rapamycin BOSS	9.42 × 10^−6^	7.99 × 10^−4^	STK11, FOXO3, FOXO1
dexamethasone BOSS	1.00 × 10^−5^	7.99 × 10^−4^	FOXO3, FOXO1, RELA
resveratrol BOSS	1.38 × 10^−5^	8.81 × 10^−4^	FOXO3, FOXO1, RELA
LY 294002 CTD 00003061	1.42 × 10^−5^	8.81 × 10^−4^	FOXO3, FOXO1, RELA
rapamycin CTD 00007350	1.83 × 10^−5^	1.02 × 10^−3^	FOXO3, FOXO1, RELA

## Data Availability

Not applicable.

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
