# Peer review of "Expression Profiles of Long Non-Coding RNAs in the Articular Cartilage of Rats Exposed to T-2 Toxin"

_ijms, 2023, doi:10.3390/ijms241813703_

Round 1

Reviewer 1 Report

In this study the authors performed RNAseq of total RNA isolated from articular cartilage of rats that were treated with T2 toxin for 4 weeks. The authors identified several lncRNA in toxin treated animals that regulate Foxo transcription factors. The authors conclude that these lncRNAs regulate autophagy. While th analysis is well performed and described the study is descriptive and does not provide any evidence showing whether and how toxin treatment changes auto-Nagy in chondrocytes. Also it is not clear how these rats were affected by 4 week toxin treatment, and especially how the joint was affected. Therefore, the study is to premature for publication.

Reviewer 2 Report

In the paper entitled  “Long noncoding RNAs expression profiles in the articular cartilage of rat exposed to T-2 toxin” authors tried to show significant correlation between autophagy process and the rat articular cartilage injury induced by T-2 toxin. In addition, they described several LncRNA (NONRATG014223.2, NONRATG012484.2, NONRATG021591.2, NONRATG024691.2 and NONRATG002808.2) and their target genes (Foxo3, Foxo1, Stk11, Hdac4 and Rela) as a key regulator factors in the process.

The manuscript is well written and easy-reading. The experiments are well designed and well presented and the methodology and statistical tests used are correct.

Although their findings may be of interest, a major revision is required to meet the standards of the journal. Many parts of the paper need to be corrected and some biological validation experiments  would be necessary to have a publishable version of this work.

More specifically, the paper has both minor and major issues.

Minor issues:

-In the introduction, the authors should introduce the concept of autophagy and also provide a brief explanation of the molecular components involved in the process. This way, the reader will have a better understanding of the rationale behind studying the relationship between cartilage injury induced by T-2 toxin and the selection of LncRNA and their target genes involved in autophagy in the results section.

-I recommend adding a simple scheme of the autophagic process for a better understanding of the results as a new figure. The figure should incorporate the following steps: Nucleation, Elongation, Formation of the mature autophagosome, Fusion, Degradation and Recycling.

-This will provide a visual representation of the different stages involved in autophagy and help readers grasp the overall process more effectively.

Major issues:

Figure 6: In this figure, validation should be performed in both control and T2 groups. Currently it is not explicit if the changes are in the control group or in the t2 group. Please show the comparison in both groups and perform the corresponding statistical tests to validate the results.

In addition, Figure 6 should also include the validation of NONRATG022809.2 ,NONRAT016242.2, NONRATG014223.2, NONRATG012484.2, NONRATG021591.2, NONRATG024691.2 and NONRATG002808.2 due to their biological significance and expression changes according to the data shown in the volcano plot (Fig. 1).

Round 2

Reviewer 1 Report

The new Figure 1 in the revised manuscript is not very convincing. To me there is no difference in the appearance of articular cartilage from untreated and toxin treated mice. Therefore, the significance of the findings are not obvious. Furthermore, the Method paper states that rats were fed on a mice diet. That seems to be be strange.

Reviewer 2 Report

In this version of the manuscript the authors include all my suggestions. I believe that in this current form, the article it is OK for the publication. 

Round 3

Reviewer 1 Report

Introduction:

Please include a statement about the importance of studying how toxins affect articular cartilage

e.g. toxins as major contributor to cartilage damage in septic arthritis.

Figure 1:

Description of the results shown in Figure 1 needs to be changed. The cartilage lesions are not shown in Figure 1. Please use different heading.

Please remove statement about thick cartilage layer and thinner cartilage layer.

Juts mention less intense safranin O staining as indication for PG loss, and increased number of empty lacunae suggesting increased chondrocyte death. In addition, more irregular arrangement of chondrocytes can be mentioned.
